# The Safety of COVID-19 Vaccinations—We Should Rethink the Policy

**DOI:** 10.3390/vaccines9070693

**Published:** 2021-06-24

**Authors:** Harald Walach, Rainer J. Klement, Wouter Aukema

**Affiliations:** 1Poznan University of the Medical Sciences, Pediatric Hospital, 60-572 Poznan, Poland; 2Department of Psychology, University of Witten/Herdecke, 58448 Witten, Germany; 3Change Health Science Institute, 10178 Berlin, Germany; 4Department of Radiation Oncology, Leopoldina Hospital, 97422 Schweinfurt, Germany; rainer_klement@gmx.de; 5Independent Data and Pattern Scientist, Brinkenbergweg 1, 7351 BD Hoenderloo, The Netherlands; wouter.aukema@gmail.com

**Keywords:** SARS-CoV2, COVID-19, vaccination, mRNA-vaccine, number needed to vaccinate, safety, side effects, adverse drug reaction, fatal side effects, EMA

## Abstract

**Background**: COVID-19 vaccines have had expedited reviews without sufficient safety data. We wanted to compare risks and benefits. **Method**: We calculated the number needed to vaccinate (NNTV) from a large Israeli field study to prevent one death. We accessed the Adverse Drug Reactions (ADR) database of the European Medicines Agency and of the Dutch National Register (lareb.nl) to extract the number of cases reporting severe side effects and the number of cases with fatal side effects. **Result**: The NNTV is between 200–700 to prevent one case of COVID-19 for the mRNA vaccine marketed by Pfizer, while the NNTV to prevent one death is between 9000 and 50,000 (95% confidence interval), with 16,000 as a point estimate. The number of cases experiencing adverse reactions has been reported to be 700 per 100,000 vaccinations. Currently, we see 16 serious side effects per 100,000 vaccinations, and the number of fatal side effects is at 4.11/100,000 vaccinations. For three deaths prevented by vaccination we have to accept two inflicted by vaccination. **Conclusions**: This lack of clear benefit should cause governments to rethink their vaccination policy.

## 1. Introduction

In the course of the SARS-CoV2 pandemic, new regulatory frameworks were put in place that allowed for the expedited review of data and admission of new vaccines without safety data [1]. Many of the new vaccines use completely new technologies that have never been used in humans before. The rationale for this action was that the pandemic was such a ubiquitous and dangerous threat that it warrants exceptional measures. In due course, the vaccination campaign against SARS-CoV2 has started. To date (18 June 2021), roughly 304.5 million vaccination doses have been administered in the EU (https://qap.ecdc.europa.eu/public/extensions/COVID-19/vaccine-tracker.html#distribution-tab (accessed on 18 June 2021)), mostly the vector vaccination product developed by the Oxford vaccination group and marketed by AstraZeneca, Vaxzevria [2] (approximately 25% coverage in the EU), the RNA vaccination product of BioNTec marketed by Pfizer, Comirnaty [3,4] (approximately 60%), and the mRNA vaccination product developed by Moderna [5] (approximately 10%). Others account for only around 5% of all vaccinations. As these vaccines have never been tested for their safety in prospective post-marketing surveillance studies, we thought it useful to determine the effectiveness of the vaccines and to compare them with the costs in terms of side effects.

## 2. Methods

We used a large Israeli field study [6] that involved approximately one million persons and the data reported therein to calculate the number needed to vaccinate (NNTV) to prevent one case of SARS-CoV2 infection and to prevent one death caused by COVID-19. In addition, we used the most prominent trial data from regulatory phase 3 trials to assess the NNTV [4,5,7]. The NNTV is the reciprocal of the absolute risk difference between risk in the treated group and in the control group, expressed as decimals. To give an artificial example: An absolute risk difference between a risk of 0.8 in the control group and a risk of 0.3 in the treated group would result in an absolute risk difference of 0.5; thus, the number needed to treat or the NNTV would be 1/0.5 = 2. This is the clinical effectiveness of the vaccine.

We checked the Adverse Drug Reaction (ADR) database of the European Medicine Agency (EMA: http://www.adrreports.eu/en/search_subst.html#, accessed on 28 May 2021; the COVID-19 vaccines are accessible under “C” in the index). Looking up the number of single cases with side effects reported for the three most widely used vaccines (Comirnaty by BioNTech/Pfizer, the vector vaccination product Vaxzevria marketed by AstraZeneca, and the mRNA vaccine by Moderna) by country, we discovered that the reporting of side effects varies by a factor of 47 (Figure 1). While the European average is 127 individual case safety reports (ICSRs), i.e., cases with side effect reports, per 100,000 vaccinations, the Dutch authorities have registered 701 reports per 100,000 vaccinations, while Poland has registered only 15 ISCRs per 100,000 vaccinations. Assuming that this difference is not due to differential national susceptibility to vaccination side effects, but due to different national reporting standards, we decided to use the data of the Dutch national register (https://www.lareb.nl/coronameldingen; accessed on 29 May 2021) to gauge the number of severe and fatal side effects per 100,000 vaccinations. We compare these quantities to the NNTV to prevent one clinical case of and one fatality by COVID-19.

## 3. Results

Cunningham was the first to point out the high NNTV in a non-peer-reviewed comment: Around 256 persons needed to vaccinate with the Pfizer vaccine to prevent one case [8]. A recent large field study in Israel with more than a million participants [6], where Comirnaty, the mRNA vaccination product marketed by Pfizer, was applied allowed us to calculate the figure more precisely. Table 1 presents the data of this study based on matched pairs, using propensity score matching with a large number of baseline variables, in which both the vaccinated and unvaccinated persons were still at risk at the beginning of a specified period [6]. We mainly used the estimates from Table 1, because they are likely closer to real life and derived from the largest field study to date. However, we also report the data from the phase 3 trials conducted for obtaining regulatory approval in Table 2 and used them for a sensitivity analysis.

It should be noted that in the Israeli field study, the cumulative incidence of the infection, visible in the control group after seven days, was low (Kaplan–Meier estimate <0.5%; Figure 2 in Dagan et al.’s work [6]) and remained below 3% after six weeks. In the other studies, the incidence figures after three to six weeks in the placebo groups were similarly low, between 0.85% and 1.8%. The absolute infection risk reductions given by Dagan et al. [6] translated into an NNTV of 486 (95% CI, 417–589) two to three weeks after the first dose, or 117 (90–161) after the second dose until the end of follow-up to prevent one documented case (Table 1). Estimates of NNTV to prevent CoV2 infection from the phase 3 trials of the most widely used vaccination products [3,4,5] were between 61 (Moderna) and 123 (Table 2) and were estimated to be 256 by Cunningham [8]. However, it should also be noted that the outcome “Documented infection” in Table 1 refers to CoV2 infection as defined by a positive PCR test, i.e., without considering false-positive results [10], so that the outcome “symptomatic illness” may better reflect vaccine effectiveness. If clinically symptomatic COVID-19 until the end of follow-up was used as an outcome, the NNTV was estimated as 217 (95% CI, 154–304).

In the Israeli field study, 4460 persons in the vaccination group became infected during the study period and nine persons died, translating into an infection fatality rate (IFR) of 0.2% in the vaccination group. In the control group, 6100 became infected and 32 died, resulting in an IFR of 0.5%, which is within the range found by a review [11].

Using the data from Table 1, we calculated the absolute risk difference to be 0.00006 (ARD for preventing one death after three to four weeks), which translates into an NNTV of 16,667. The 95% confidence interval spanned the range from 9000 to 50,000. Thus, between 9000 and 50,000 people need to be vaccinated, with a point-estimate of roughly 16,000, to prevent one COVID-19-related death.

For the other studies listed in Table 2, in the case that positive infection was the outcome [7], we calculated the NNTV to prevent one death using the IFR estimate of 0.5%; in the case that clinically positive COVID-19 was the outcome [4,5], we used the case fatality rate estimated as the number of worldwide COVID-19 cases divided by COVID-19 related deaths, which was 2% (https://www.worldometers.info/coronavirus/ (accessed on 29 May 2021)). In the case of the Sputnik vaccine, one would thus have to vaccinate 22,000 people to prevent one death. In the case of the Moderna vaccine, one would have to vaccinate 3050 people to prevent one death. In the case of Comirnaty, the Pfizer vaccine, 6150 vaccinated people would prevent one death, although using the figure by Cunningham [8], it would be 12,300 vaccinations to prevent one death.

The side effects data reported in the Dutch register (www.lareb.nl/coronameldingen (accessed on 27 May 2021)) are given in Table 3.

Thus, we need to accept that around 16 cases will develop severe adverse reactions from COVID-19 vaccines per 100,000 vaccinations delivered, and approximately four people will die from the consequences of being vaccinated per 100,000 vaccinations delivered. Adopting the point estimate of NNTV = 16,000 (95% CI, 9000–50,000) to prevent one COVID-19-related death, for every six (95% CI, 2–11) deaths prevented by vaccination, we may incur four deaths as a consequence of or associated with the vaccination. Simply put: As we prevent three deaths by vaccinating, we incur two deaths.

The risk–benefit ratio looks better if we accept the stronger effect sizes from the phase 3 trials. Using Cunningham’s estimate of NNTV = 12,300, which stems from a non-peer reviewed comment, we arrived at eight deaths prevented per 100,000 vaccinations and, in the best case, 33 deaths prevented by 100,000 vaccinations. Thus, in the optimum case, we risk four deaths to prevent 33 deaths, a risk–benefit ratio of 1:8. The risk–benefit ratio in terms of deaths prevented and deaths incurred thus ranges from 2:3 to 1:8, although real-life data also support ratios as high as 2:1, i.e., twice as high a risk of death from the vaccination compared to COVID-19, within the 95% confidence limit.

## 4. Discussion

The COVID-19 vaccines are immunologically effective and can—according to the publications—prevent infections, morbidity, and mortality associated with SARS-CoV2; however, they incur costs. Apart from the economic costs, there are comparatively high rates of side effects and fatalities. The current figure is around four fatalities per 100,000 vaccinations, as documented by the most thorough European documentation system, the Dutch side effects register (lareb.nl). This tallies well with a recently conducted analysis of the U.S. vaccine adverse reactions reporting system, which found 3.4 fatalities per 100,000 vaccinations, mostly with the Comirnaty (Pfizer) and Moderna vaccines [12].

Is this a few or many? This is difficult to say, and the answer is dependent on one’s view of how severe the pandemic is and whether the common assumption that there is hardly any innate immunological defense or cross-reactional immunity is true. Some argue that we can assume cross-reactivity of antibodies to conventional coronaviruses in 30–50% of the population [13,14,15,16]. This might explain why children and younger people are rarely afflicted by SARS-CoV2 [17,18,19]. An innate immune reaction is difficult to gauge. Thus, low seroprevalence figures [20,21,22] may not only reflect a lack of herd immunity, but also a mix of undetected cross-reactivity of antibodies to other coronaviruses, as well as clearing of infection by innate immunity.

However, one should consider the simple legal fact that a death associated with a vaccination is different in kind and legal status from a death suffered as a consequence of an incidental infection.

Our data should be viewed in the light of its inherent limitations:

The study which we used to gauge the NNTV was a single field study, even though it is the largest to date. The other data stem from regulatory trials that were not designed to detect maximum effects. The field study was somewhat specific to the situation in Israel, and studies in other countries and other populations or other post-marketing surveillance studies might reveal more beneficial clinical effect sizes when the prevalence of the infection is higher. This field study also suffered from some problems, as a lot of cases were censored due to unknown reasons, presumably due to a loss to follow-up. However, the regulatory studies compensate for some of the weaknesses, and thereby generate a somewhat more beneficial risk–benefit ratio.

The ADR database of the EMA collects reports of different kinds, by doctors, patients, and authorities. We observed (Figure 1) that the reporting standards vary hugely across countries. It might be necessary for the EMA and for national governments to install better monitoring procedures in order to generate more reliable data. Some countries have tight reporting schemes, some report in a rather loose fashion. As we have to assume that the average number of side effects is roughly similar across countries, we would expect a similar reporting quota. However, when inspecting the reports according to countries, we can see a large variance. Our decision to use the Dutch data as a proxy for Europe was derived from this discovery. One might want to challenge this decision, but we did not find any data from other countries being more valid than those used here. Apart from this, our data tallied well with the data from the U.S. CDC vaccine adverse reporting system [12], which indirectly validates our decision.

One might argue that it is always difficult to ascertain causality in such reports. This is certainly true; however, the Dutch data, especially the fatal cases, were certified by medical specialists (https://www.lareb.nl/media/eacjg2eq/beleidsplan-2015-2019.pdf (accessed on 29 May 2021)), page 13: “*All reports received are checked for completeness and possible ambiguities. If necessary, additional information is requested from the reporting party and/or the treating doctor The report is entered into the database with all the necessary information. Side effects are coded according to the applicable (international) standards. Subsequently an individual assessment of the report is made. The reports are forwarded to the European database (Eudravigilance) and the database of the WHO Collaborating Centre for International Drug Monitoring in Uppsala. The registration holders are informed about the reports concerning their product*.”).

A recent experimental study showed that the SARS-CoV2 spike protein is sufficient to produce endothelial damage [23]. This provides a potential causal rationale for the most serious and most frequent side effects, namely, vascular problems such as thrombotic events. The vector-based COVID-19 vaccines can produce soluble spike proteins, which multiply the potential damage sites [24]. The spike protein also contains domains that may bind to cholinergic receptors, thereby compromising the cholinergic anti-inflammatory pathways, enhancing inflammatory processes [25]. A recent review listed several other potential side effects of COVID-19 mRNA vaccines that may also emerge later than in the observation periods covered here [26].

In the Israeli field study, the observation period was six weeks, and in the U.S. regulatory studies between four to six weeks, a period commonly assumed to be sufficient to see a clinical effect of a vaccine, because it would also be the time frame within which someone who was infected initially would fall ill and perhaps die. Had the observation period been longer, the clinical effect size might have increased, i.e., the NNTV could have become lower and, consequently, the ratio of benefit to harm could have increased in favor of the vaccines. However, as noted above, there is also the possibility of side effects developing with some delay and influencing the risk–benefit ratio in the opposite direction [26]. This should be studied more systematically in a long-term observational study.

Another point to consider is that initially, mainly older persons and those at risk were entered into the national vaccination programs. It is to be hoped that the tally of fatalities will become lower as a consequence of the vaccinations, as the age of those vaccinated decreases.

However, we do think that, given the data, we should not wait to see whether more fatalities accrue, but instead use the data available to study who might be at risk of suffering side effects and pursue a diligent route.

Finally, we note that from experience with reporting side effects from other drugs, only a small fraction of side effects is reported to adverse events databases [27,28]. The median underreporting can be as high as 95% [29].

Given this fact and the high number of serious side effects already reported, the current political trend to vaccinate children who are at very low risk of suffering from COVID-19 in the first place must be reconsidered.

## 5. Conclusions

The present assessment raises the question whether it would be necessary to rethink policies and use COVID-19 vaccines more sparingly and with some discretion only in those that are willing to accept the risk because they feel more at risk from the true infection than the mock infection. Perhaps it might be necessary to dampen the enthusiasm by sober facts? In our view, the EMA and national authorities should instigate a safety review into the safety database of COVID-19 vaccines and governments should carefully consider their policies in light of these data. Ideally, independent scientists should carry out thorough case reviews of the very severe cases, so that there can be evidence-based recommendations on who is likely to benefit from a SARS-CoV2 vaccination and who is in danger of suffering from side effects. Currently, our estimates show that we have to accept four fatal and 16 serious side effects per 100,000 vaccinations in order to save the lives of 2–11 individuals per 100,000 vaccinations, placing risks and benefits on the same order of magnitude.

## Figures and Tables

**Figure 1 vaccines-09-00693-f001:** Individual safety case reports in association with COVID 19 vaccines in Europe.

**Table 1 vaccines-09-00693-t001:** Risk differences and number needed to vaccinate (NNTV) to prevent one infection, one case of symptomatic illness, and one death from COVID-19. Data from Dagan et al. [6], *N* = 596,618 in each group.

	Documented Infection	Symptomatic Illness	Death from COVID-19
Period	Risk Difference (No./1000 Persons) (95% CI)	NNTV (95% CI)	Risk Difference (No./1000 Persons) (95% CI)	NNTV (95% CI)	Risk Difference (No./1000 Persons) (95% CI)	NNTV (95% CI)
14–20 days after first dose	2.06 (1.70–2.40)	486 (417–589)	1.54 (1.28–1.80)	650 (556–782)	0.03 (0.01–0.07)	33,334 (14,286–100,000)
21–27 days after first dose	2.31 (1.96–2.69)	433 (372–511)	1.34 (1.09–1.62)	747 (618–918)	0.06 (0.02–0.11)	16,667 (9091–50,000)
7 days after second dose to end of follow-up	8.58 (6.22–11.18)	117 (90–161)	4.61 (3.29–6.53)	217 (154–304)	NA	NA

Data taken from Table 2 in Dagan et al.’s work. NNTV = 1/risk difference.

**Table 2 vaccines-09-00693-t002:** Number needed to vaccinate (NNTV) calculated from pivotal phase 3 regulatory trials of the SARS-CoV2 mRNA vaccines of Moderna, BioNTech/Pfizer, and Sputnik (the vector vaccine of Astra-Zeneca is not contained here, as the study [9] was active-controlled and not placebo-controlled).

Vaccine	*N* Participants Vaccine Group	*N* Participants Placebo Group	CoV2 Positive End of Trial Vaccine Group	CoV2 Positive End of Trial Placebo Group	Absolute Risk Difference (ARD)	Number Needed to Vaccinate 1/ARR
Moderna [5] ^$^	15,181(14,550 *)	15,170 (14,598 *)	19 (0.13%) ^1^	269 (1.77%) ^1^	0.0165	61
Comirnaty (BioNTech/Pfizer) [4] ^$^	18,860	18,846	8 (0.042%) ^2^	162 (0.86%) ^2^	0.00817	123
Sputnik V [7] ^§^	14,964	4902	13 (0.087%) **^,3^	47 (1%) **^,3^	0.0091	110

* Modified intention to treat-population—basis for calculation; ** taken from the publication because of slightly different case numbers; $ outcome was a symptomatic COVID-19 case; § outcome was a confirmed infection by PCR-test; ^1^ after 6 weeks; ^2^ after 4 weeks; ^3^ after 3 weeks.

**Table 3 vaccines-09-00693-t003:** Individual case safety reports for the most widely distributed COVID-19 vaccines according to the Dutch side effects register (www.lareb.nl/coronameldingen (accessed on 29 May 2021)), the absolute numbers per vaccine, and standardization per 100,000 vaccinations.

	General Number of Reports (1)	Serious Side Effects (1)	Deaths (2)	Number of Vaccinations According to (3)	Number of Vaccinations According to ECDC (4)
Comirnaty (Pfizer)	21,321	864	280	5,946,031	6,004,808
Moderna	6390	114	35	531,449	540,862
Vaxzevria (AstraZeneca)	29,865	411	31	1,837,407	1,852,996
Janssen	2596	7	-	142,069	143,525
Unknown	129	15	5	-	540
Total	60,301	1.411	351	8,456,956	8,542,731
**Per 100,000 vaccinations according to Dutch data**	**713.03**	**16.68**	**4.15**		
**Per 100,000 vaccinations according to ECDC**	**705.87**	**16.52**	**4.11**		

(1) https://www.lareb.nl/coronameldingen. (2) https://www.lareb.nl/pages/update-van-bijwerkingen. (3) https://coronadashboard.rijksoverheid.nl/landelijk/vaccinaties. (4) https://www.ecdc.europa.eu/en/publications-data/data-covid-19-vaccination-eu-eea. All sites accessed on 27 May 2021. The Dutch government reported two numbers; we took the calculated amounts.

## Data Availability

Documentation on how to extract information from the line listings of the ADR-database of the EMA, SQL scripts, and graphical representations is available at http://www.aukema.org/2021/04/analysis-of-icsr-reports-at-emaeuropaeu.html (accessed on 22 June 2021).

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
