# Peer review of "RETRACTED: The Safety of COVID-19 Vaccinations—We Should Rethink the Policy"

_vaccines, 2021, doi:10.3390/vaccines9070693_

Round 1

Reviewer 1 Report

The manuscript by Walach et al. provides an urgently awaited analysis of the benefits and risks of COVID-19 vaccination under real-life field conditions, based on large sets of statistical data that are only becoming available these days. Even though some of these underlying publically available numbers may carry a certain intrinsic bias (due to reporting of side effects, and/or priority of vaccination), the present analysis is performed responsibly (e.g. with plausible and convincing arguments as to why the Dutch EMA data set was chosen) and without methodological flaws, and the results are interpreted with the necessary caveats. Interestingly, this up-to-date comparison of the NNTV (Number Needed to Vaccinate) and the adverse drug reactions collected by the EMA (European Medicine Agency) shows that the risk of infection by COVID-19 in terms of severe side effects and death is comparable – within roughly an order of magnitude – with the risks incurred by vaccination with the prevalent products in Europe (Pfizer, Moderna, AstraZeneca). This finding is a timely contribution to help with the overall risk-benefit assessment, both, at the level of governments and doctors in charge of health care, as well as each person – especially seniors and children – who may wish to choose wisely whether to take a vaccination depending on their individual risk profile.

Some minor points should be corrected before publication:

Line 1, 14, and throughout: capitalize “COVID-19”

Line 59: The authors may like to emphasize that they did not just add up all (multiple) side effects in the ADR database, but that they refer strictly to the number of reported cases.

Line 89: hyphenate “active-controlled”

Table 2, columns 4 and 5: add a footnote on the observation period of these studies

Table 3, 1st column: zero missing in “per 1oo,oo0 vaccinations”, and use comma as separator

Table 3, footnote 2: link does not open

Table 3, footnote 4: letter missing in “… Dutch government reports two numberS …”

Line 142: it should be “per 100,000 VACCINATIONS DELIVERED” (as people ought to get vaccinated twice, though this is not significant within the 6 week observation period).

Lines 144-145: use consistent format for small numbers in this sentence, i.e. spell out “for every SIX”

Lines 150-152: reverse the order of numbers to reflect the risk-to-benefit ratio from the point of view of vaccination, i.e. “we risk 4 deaths to prevent 33 deaths”.

Line 177-178: reformulate ambiguous singular-plural “The DATA which we used to gauge the NNTV ARE BASED ON a single, albeit the largest field study to date, AND ON regulatory trials that are not designed…”

Lines 199-200: duplication of sentence

Line 234: state more clearly “THE PRESENT ASSESSMENT RAISES THE QUESTION WHETHER IT WOULD BE useful to rethink policies and use COVID-19 vaccines MORE sparingly…”

Line 240: insert comma “… severe cases, so that …”

Author Response

We have accomodated all suggestions of the reviewer. See attached point-by-point reply

Reviewer 2 Report

The manuscript by Walach et al is very important and should be published urgently. Please update the data about the number of vaccinations in EU. Please avoid the use of the names "Pfizer", "Moderna" etc and prefer the official names of the vaccines.

Discussion: Within the sentence: "The COVID-19 vaccines are immunologicaly effective and can prevent..." the authors should include the phrase "according to the publications". 

The authors are describing literature about the potential toxic effect of SPIKE. Please add the reference by Farsalinos et al, IJMS (https://www.mdpi.com/1422-0067/21/16/5807) describing the snake toxin-like epitope of the SPIKE protein and discuss it.

Author Response

Thank you for your appreciative comments. We have done the following:

  • included the information about the Farsalinos-publication and discussed it in the Discussion
  • added the suggested phrase "according to the publications"
  • added the brandnames for the Pfizer and Astra-Zeneca vaccines in the tables and the text and used the same convention as the ECDC Dashboard which uses Moderna and Sputnik (and not their brandnames)

We have  updated the vaccination data in the introductory part. But we have not done that in the rest of the paper, as this would only make sense if we updated the full database. This is time consuming and would delay the publication. We think it is more important to get this information out as quickly as possible. We will update the information at a later point in time with more diligent analyses according to age groups and gender, etc. We hope for your understanding.

Reviewer 3 Report

The manuscript ‘The Safety of Covid-19 Vaccinations – Should we Rethink the Policy’ by Walach and colleagues has been reviewed.

In their paper, Walach and co-workers tried to determine the effectiveness of the currently available vaccines by comparing their SARS-CoV-2 protecting effect with the side effects.

The manuscript is well-written and due to the current pandemic it is of certain interest.

Nevertheless, it is unclear why the study has been performed by comparing NNTV obtained from Israeli study with the side effects reported in the European Medicines Agency and of the Dutch National Register. In my opinion, due to economical and social differences, the data should be calculated using the local register.

As clearly reported by The BMJ editor (and in the reference), the opinion reported by Cunningham is an online comment by a third party and not a reviewed paper. This should be clearly reported in the discussion.

The literature is adequate (even if should be thoroughly revised), but several cited texts are not peer-reviewed papers, and this should be clearly stated.

Minor concerns:

The manuscript should be thoroughly revised to correct misprinting (eg table 1, line 78, 108, 109, 111, 142, 151 etc)

Literature should be revised including all the fundamental information (year, issue, pages) and the style properly uniformed (see ref. 2, 3, 6, 7, 17)

The quality of Figure 1 should be improved

In my opinion, the manuscript should be accepted after major revisions noted.

Author Response

Please see the attachment. A native speaker has chequed the revised text and made some very minor adjustments.

Round 2

Reviewer 3 Report

The paper should be accepted as it is